# A Taxonomy-Agnostic Approach to Targeted Microbiome Therapeutics—Leveraging Principles of Systems Biology

**DOI:** 10.3390/pathogens12020238

**Published:** 2023-02-02

**Authors:** Kyle D. Brumfield, Paul Cox, James Geyer, Julius Goepp

**Affiliations:** 1Maryland Pathogen Research Institute, University of Maryland, College Park, MD 20742, USA; 2Institute for Advanced Computer Studies, University of Maryland, College Park, MD 20742, USA; 3Evimero, Tuscaloosa, AL 35406, USA; 4Institute for Rural Health Research, College of Community Health Science, University of Alabama, Tuscaloosa, AL 35487, USA

**Keywords:** microbiome, dysbiosis, systems biology, targeted therapeutics, FMT, phage therapy

## Abstract

The study of human microbiomes has yielded insights into basic science, and applied therapeutics are emerging. However, conflicting definitions of what microbiomes are and how they affect the health of the “host” are less understood. A major impediment towards systematic design, discovery, and implementation of targeted microbiome therapeutics is the continued reliance on taxonomic indicators to define microbiomes in health and disease. Such reliance often confounds analyses, potentially suggesting associations where there are none, and conversely failing to identify significant, causal relationships. This review article discusses recent discoveries pointing towards a molecular understanding of microbiome “dysbiosis” and away from a purely taxonomic approach. We highlight the growing role of systems biological principles in the complex interrelationships between the gut microbiome and host cells, and review current approaches commonly used in targeted microbiome therapeutics, including fecal microbial transplant, bacteriophage therapies, and the use of metabolic toxins to selectively eliminate specific taxa from dysbiotic microbiomes. These approaches, however, remain wholly or partially dependent on the bacterial taxa involved in dysbiosis, and therefore may not capitalize fully on many therapeutic opportunities presented at the bioactive molecular level. New technologies capable of addressing microbiome-associated diseases as molecular problems, if solved, will open possibilities of new classes and categories of targeted microbiome therapeutics aimed, in principle, at all dysbiosis-driven disorders.

## 1. Background and Introduction

Interest in the human microbiome and its implications for health maintenance, disease prevention, and disease treatment has skyrocketed in the past decade as new molecular techniques permit increasingly high-resolution views of the microbial populations that live in and on all living things [1]. In humans, the gut microbiome—the ecosystem of microorganisms present throughout the entire gastrointestinal tract—has been dubbed a “new organ,” in recognition of the multiple cellular and biochemical interactions both within the organ itself and with the cells of the “host;” the collective collection of microbial and human cells is commonly now referred to as the “holobiont [2,3,4].” In fact, human gut microbiota are found in higher proportion than cells of the human body [5]; the total number of genes coded for by gut microbiota is even greater in comparison to the number of genes encoded by the human genome [6], producing bioactive molecules at scales dwarfed by those produced by human cells [7]. Many metagenomic investigations have aimed to understand the metabolism of human gut microorganisms, especially nonculturable microbiota, and their relationship with maintaining homeostasis [8]. In addition to supporting digestion, the gut microbiome contributes to host cell function, including production of metabolites that cannot be manufactured by human cells, prime examples being B vitamins [9], thiamine and riboflavin [10], and vitamin K [11]. Moreover, gut microbiota composition and function play important roles in regulating the immune system and sustaining resistance to pathogen colonization [8]. For example, microbe-mediated actions help sustain the integrity of the essential gut permeability barrier that maintains separation between the mostly microbial contents of the gut and the primarily human cells comprising the body [12]. Dekaboruah et al. [13] have detailed numerous further examples of the host benefits of this complex symbiotic relationship. 

Today, there is widespread agreement that well-balanced gut microbial ecosystems are stable (vary little over time), resilient (capable of returning to a pre-existing equilibrium state), and resistant (to colonization by pathogens), while disruptions of human gut microbial communities (“dysbiosis”) have close associations with many chronic and non-communicable disorders [2]; increasingly, mechanistic explanations for these associations are becoming clear [14,15,16,17]. Stable, resilient, and colonization-resistant microbiomes, such as macroscopic ecosystems, can tolerate occasional small environmental disruptions (e.g., dietary indiscretions, short infrequent courses of antibiotics), and therefore act in a host-protective fashion [2]. An important and often-overlooked factor, however, is that prolonged and sizable disruptions (e.g., chronic caloric excess, frequent and prolonged antibiotic treatments) can establish a state of dysbiosis that itself demonstrates stability, resilience, and resistance from the standpoint of the microbiome alone, but one that does not contribute to homeostasis (or health) of the holobiont, and may therefore become disease-permissive or even disease-promoting [2,18]. 

Figure 1 shows a diagram of the impacts of dysbiosis on systems in the host, or holobiont.

Such environmental disruptions often involve dysbiosis of autochthonous (naturally resident) microbiota followed by “blooms” of opportunistic pathogens (sometimes called “pathobionts”), which can quickly become dominant as a result of environmental disruptions, such as poor diet, frequent antibiotic courses, or environmental toxins [19,20,21,22]. Disruptions of microbiome composition are found at every taxonomic level, with some of the earliest observations relating to the balance of phyla *Firmicutes* and *Bacteroidetes*, while more recently, a strain-level dysbiosis has been described [23]. It is now well established, for example, that members of the Family *Enterobacteriaceae*, while present in healthy people’s gut microbiomes at less than 1% of the total microbiota, can bloom (and potentially attain “monodominance [24,25,26,27]”) in inflamed guts [28]. Single species may also bloom in response to such perturbations, especially prolonged antibiotic use, as seen in *Clostridiodes difficile* enteritis [29].

It is important to note in this context that the seeds of dysbiosis may be planted very early in life; the first 1000 days of life (roughly from birth to two years of age) is known to be crucial for development of immune, endocrine, metabolic, and other developmental pathways in offspring [30]. Because the neonatal microbiome is nearly entirely obtained from the maternal gut microbiome, dysbiosis may be intergenerational, and can perpetuate physiological impairments into successive generations [30]. Recent findings also show that the horizontal, person-to-person transmission of gut and oral microbiomes is important in shaping individual microbiota [31].

## 2. Defining “Microbiome” Matters in Targeted Therapeutics Development

Definitions of “microbiome” have varied, from an early “census-like” notion wherein the human microbiome is the “collection of all the microorganisms living in association with the human body [5,13,32]”, to a more nuanced and inclusive view that a microbiome consists of “The genes and genomes of the microbiota, as well as the products of the microbiota and the host environment [32,33]”.

It is now established that culture-dependent methods for detecting and enumerating taxa of the human gut microbiome can introduce bias, since the vast majority of prokaryotic genospecies remain uncultured [34]. That is, genomes of uncultured microorganisms have potential to encode novel metabolites and metabolic processes. Metagenomic (DNA) and metatranscriptomic (RNA) sequencing effectively obviates the need to isolate and culture microorganisms by utilizing genetic material of a sample to accurately profile microbiota and identify functional gene composition [1,35]. Polymerase chain reaction (PCR) is a fundamental method commonly employed for taxonomic identification, by amplifying variant regions in macromolecules conserved among certain taxa [36]. PCR-based metagenomics is now a common method used to evaluate microbial species diversity based on sequence composition. Specifically, PCR amplification of 16S ribosomal (rRNA) genes, occurring in one or more copies in most bacterial and archaeal genomes, is routinely employed to amplify hypervariable regions of the 16S rRNA gene to infer taxonomic identification by bioinformatic alignment against various sequence databases. Per contra, instead of targeting specific genomic markers, whole genome shotgun metagenomic sequencing, whereby total DNA is sheared into fragments that can be independently sequenced and aligned, allows researchers to profile all the genes in all microorganisms present in uncultured microbial communities [37], using them not only for taxonomic identification but also for discerning potential of functional profiles. Similarly, the use of metatranscriptomics in microbiome research has allowed researchers to gain insight into genes that are actively expressed, detecting functional changes that dictate contextual fluctuations, microbiome-host interactions, and functional alterations associated with conversion of a microbiome towards dysbiosis [38]. 

While DNA/RNA sequencing of the human microbiome has expanded our understanding of host-microbe interactions, low-abundant and previously undescribed species have largely been overlooked, while their contributions to molecular-level dysbiosis and diseased phenotype may be substantial [39]. Researchers are now beginning to understand the key players in the human gut microbiome. However, a considerable portion of the microbiome is still considered “dark matter,” and the field has not reached consensus on the profile of a “healthy microbiome” as many differences arise through a combination of environmental, genetic, and lifestyle factors [40]. Hence, more research is needed to elucidate physiological and biological mechanisms, including virulence and antimicrobial determinants, and taxonomic biomarkers for diagnosis and treatment, in context of the transition of a healthy microbiome to a state of dysbiosis.

The vast majority of human microbiome research has focused on bacterial members of the gut ecosystem, primarily because bacteria are the dominant domain profiled with the use of conventional methods. However, it should be noted that strong evidence exists for similar, if quantitatively different, contributions of viral and fungal communities, i.e., the virome and mycobiome, respectively [41,42]. As these fields continue to mature, molecular mediators of dysbiosis can be expected to be identified, and potential therapeutics can be developed accordingly. 

The foregoing discussion highlighting the importance of molecular-level contributors to dysbiosis helps to explain why adherence to a strict taxon-driven “census” view of microbiomes has at times confounded studies of microbiome-disease associations, with multiple studies of the same disease yielding dissimilar profiles of microbiome taxonomic composition [43]. Indeed, such an approach has contributed to the so-called “Anna Karenina Effect [44,45];” just as Tolstoy’s novel opens with the lines, “Happy families are all alike; every unhappy family is unhappy in its own way,” in the microbiome/disease setting, “Dysbiotic individuals vary more in microbial community composition than healthy individuals [45].” 

The application of the principles of systems biology [46], which explores how individual components (cells, molecules, pathways) of biological systems interact and ultimately give rise to an observed phenotype [47], offers a further solution to the Anna Karenina effect. Shifting the analytical framework away from taxonomic definitions, and towards parameters that measure actual molecular interactions between and among microbial and human cells, yields associations many-fold stronger than those identified using taxonomy alone. 

A recent study by Tierney et al. [43], for example, showed that gene-level analyses of microbiome-disease relationships provided more robust associations with several microbiome-associated diseases (MADs) than taxonomic-level analyses. At a still-higher level of resolution, bioactive microbial proteins were shown to be differentially enriched in inflammatory bowel disease (IBD) patients, many of which were carried by multiple individual genera. [7] Indeed, in some cases in this analysis, proteins that were both enriched and depleted in IBD patients were found within the same species (specifically *Ruminococcus gnavus* and *Faecalibacterium prausnitzii*). Functionally, bacterial pilin proteins (typically carried by *Proteobacteria*) were among those differentially enriched in IBD patients [7]; pilins are of crucial importance for many of these bacterial opportunistic pathogens (or “pathobionts”), mediating their attachment to human intestinal epithelial cell membranes and facilitating the transmission of toxins and other metabolites that contribute to increased inflammation, loss of intestinal barrier integrity, and genomic damage [7,48,49]. Similarly, amyloid fibers called curli, important biofilm components, are produced by multiple members of the Family *Enterobacteriaceae*, and may contribute to disease phenotype in several gut and extraintestinal disorders, including some neurodegenerative diseases [16,50,51,52,53,54]. These findings are consistent with the “insurance hypothesis” notion that the biological function of a microbial community can be maintained even in the absence of specific taxa, so long as others can provide that function [55]. 

In short, a microbiome in health embodies a stable, pro-homeostasis molecular milieu (Figure 2), while a dysbiotic microbiome can be viewed as a state of ecological “molecular pollution,” wherein multiple taxa contribute to a disease phenotype in a MAD, via their collective output of bioactive proteins (Figure 3). In such a framework, it becomes possible to identify with considerable confidence specific bioactive proteins—independent of their taxonomic origin—whose reduction within the system will yield a favorable modification of the molecular milieu, potentially resulting in a return to a phenotype free of the particular MAD in question.

Viewing dysbiosis from the molecular, rather than the purely taxonomic, standpoint opens the possibility of developing highly targeted microbiome therapeutics that are taxonomy-agnostic. This approach may avoid the Anna Karenina problem and permit the development of more holistic, systems biology-based therapeutics. 

In the following section, we examine some of the existing and new approaches to targeted microbiome therapeutics, both those that depend on taxonomic community structure and those independent of taxonomy.

## 3. Targeted Microbiome Therapeutics: Opportunities and Challenges

Several different approaches have been taken to achieve the goal of targeted microbiome therapeutics, which may also be identified in literature as “precision editing” or “reprogramming” of a dysbiotic microbiome [15,56,57,58,59,60,61,62,63,64,65]. These approaches are outlined below. We note here that a discussion of probiotics and live biotherapeutics (LBPs) as microbiome therapies are beyond the scope of this article. 

### 3.1. Fecal Microbiota Transplantation (FMT)

FMT is among the oldest approaches to microbiome therapeutics; essentially, the goal is to directly change the recipient’s gut microbial composition by means of administration of fecal matter (or purified fecal microbiota) from a presumably healthy donor to resolve a state of dysbiosis [66]. FMT has proved to be highly effective so far in only one specific condition: *Clostridiodes difficile* (formerly *Clostridium difficile*) diarrhea [66], but it is also showing promise in treating other disorders related to dysbiosis, including IBD, certain autoimmune disorders, Alzheimer’s and Parkinson’s diseases [67,68,69], autoimmune diseases [70], type 2 diabetes [71], and many others [66,72,73]. 

Experimental FMT in animal models can be an effective means of establishing that some features of dysbiosis are contributory to specific diseases. Many fruitful lines of research have been initiated, for example, by transplanting fecal matter from a diseased, dysbiotic animal into a healthy one and observing recapitulation of the disease phenotype [74,75,76]. The converse is true as well, as shown by studies demonstrating relief of both dysbiosis and disease phenotype by FMT from a healthy to a diseased donor (if the phenotype changes from “diseased” to “not diseased” a contribution is inferred, and can provide some insights into relevant disease mechanisms) [77]. Thus, FMT plays an important research role in demonstrating that a disease is (a) causally related to dysbiosis, and (b) responsive to a non-specific resolution of that dysbiotic state.

Recently, the first fecal microbiota product was approved by the US FDA, developed for prevention of recurrent *Clostridiodes difficile*-induced chronic diarrhea [78]. In data from six placebo-controlled studies involving 1061 subjects, infection was cleared in 70% of drug recipients and in 58% of placebo subjects, corresponding to a relative reduction of recurrence of 29.4% compared with placebo [79,80]. Several other FMT-related drugs are currently under evaluation by FDA and other regulatory agencies [81,82].

FMT is the ultimate taxonomy-agnostic microbiome therapeutic, in that no attempt is made to remove specific species/strains of bacteria or any other constituent of the microbiota; indeed, FMT’s success may be in part attributed to the replacement of viral and fungal elements [83].

FMT, however, has multiple limitations. FMT cannot be considered a targeted microbiome therapy, nor one that incorporates any systems biology principles. Rather, it is largely a “black box” approach that essentially aims to replace both taxonomic and molecular disruptions of dysbiosis without specific targets [84]. This potentially restricts its applicability as a rational and reproducible therapeutic approach [85]. Further limitations of FMT are related to multiple practical aspects of production, selection and standardization of donors, colonization resistance, and, of course, complexities of fecal microbiome profiling along with the potential for transmitting pathogenic agents and/or antibiotic resistant microbes [77,86,87]. An additional concerning limitation is the risk of transmitting an undesirable microbiome-driven phenotype such as obesity, in essence trading one MAD for another [88]. 

Figure 4 shows a schematic summary of the pros and cons of FMT.

### 3.2. Bacteriophage Therapy

Bacteriophage viruses (“phages”) were first proposed in the early 20th century, prior to the discovery of antibiotics, as a means of controlling or eliminating infectious bacteria [89,90]. After falling out of favor with the advent of antibiotics, phage therapy is experiencing a renaissance in the treatment of antibiotic-resistant bacterial infections [91], and now in the field of microbiome therapeutics [89,91,92]. 

Phages are highly species- and often strain-specific viruses that rapidly replicate within, and then destroy target bacterial cells [91,92,93]. Phages endogenous to gut microbiomes are increasingly recognized as important dynamic factors in overall microbiome composition and function; indeed, it has been suggested that some of the microbiome composition changes seen in IBD, for example, are driven at least in part by “blooms” of endogenous phages [92]. This is an area of microbiome science that remains in its infancy, but such deleterious phages may ultimately themselves represent targets for selective therapeutic microbiome modification [41,94].

For the purposes of this review, “phage therapy” refers to the administration of phage viruses to selectively reduce populations of pathobionts known to promote MADs. Preclinical studies have demonstrated successful resolution of dysbiotic microbiomes and improvements in phenotype in alcoholic liver disease [95], intestinal inflammation and IBD [74,96], colorectal cancer, and others [74,97,98,99,100].

Phage therapies offer several key advantages in the quest for targeted microbiome therapeutics. From a research perspective, experimental phage applications can help in understanding the complex networks of bacterial contributions to MADs, even permitting identification of primary pathobiont “driver” strains and secondary “passenger” strains, in which altered abundance arises from disease-related processes, and whose alterations may be less effective in changing disease phenotype [74,101,102].

By selectively infecting and killing their target bacterial hosts, phages promise to entirely eradicate populations of pathobionts, reducing their impact on host tissues and ultimately phenotype [92,93]. Phage host-specificity is determined by specific bacterial cell-binding structures that differ according to target hosts [103]. Their high host-specificity means that they can leave uninvolved or beneficial microorganisms largely intact, limiting off-target effects [92,93]. Furthermore, the rational selection of phages and target bacteria may permit development of phage therapeutics that can overcome development of individual bacterial resistance to specific phages, especially when various phages operate by different mechanisms in attacking their targets [74,104,105,106].

A recent study highlighted many of the promises of rationally designed phage therapeutics in MADs, specifically in IBD [74]. Using metagenomic techniques, researchers identified a clade of strains of *Klebsiella pneumoniae* that were strongly associated with IBD disease exacerbation and severity; through experimental FMT of human IBD-associated *K. pneumonia* strains into mice, they were able to demonstrate microbiome-associated increased inflammation in recipient animals [74]. Subsequently, they generated a “cocktail” of five lytic phages known to lyse and destroy their target bacteria by different mechanisms, providing the requisite overlap in function to avoid resistance to any single phage. Experiments in colitis-prone mice showed that administration of this phage cocktail suppressed the offending *K. pneumoniae* strains, reducing the gut inflammatory response and disease severity as predicted. Further study in human volunteers demonstrated survival of the phages through the upper gastrointestinal tract and their viability in the colon, where the target bacteria are found. 

These findings open the door to a form of “sculpting” of a disease-prone dysbiotic microbiome in humans, using phage therapeutic techniques, and provide a proof-of-principle that such selective microbiome modulation can have a direct beneficial effect in a chronic, human, noncommunicable disease, IBD, while raising the real possibility that similar interventions may arise in the cases of other MADs [74,96]. Phage therapies, then, may be seen as highly targeted approaches to dysbiosis at the taxonomic level.

Limitations of microbiome phage therapies remain, however. While bacterial resistance can be reduced by careful selection of therapeutic phage combinations, we know from long experience with small-molecule antibiotics that bacteria can rapidly evolve strategies to overcome even the most carefully designed, mechanistically overlapping therapies [107]. While few data exist in microbiomes, experience in complex macroscopic ecosystems suggests a real risk of unintended consequences produced by introduction or translocation of predators (the approximate equivalent of lytic phages in microbiomes), and suggest that suppression, rather than eradication, may sustain desired function with fewer long-term risks [108].

One additional limitation applies to phage therapeutics in a microbiome setting, namely the vital emerging roles of non-bacterial drivers of dysbiosis. These include viruses other than phages, which are increasingly recognized as important in maintaining gut ecological balance and contributors to dysbiosis in disease [109,110,111,112], as well as fungi and protozoa [113,114,115,116,117]. Phages targeting bacteria exclusively are unable to directly affect these organisms of emerging importance. Furthermore, while phage therapeutics have been shown to indirectly affect bacterial metabolites, by their nature phages are unable to directly block or neutralize bacterial metabolites and virulence factors of the kind now recognized as important players in a dysbiotic molecular milieu [93] Finally, phage therapeutics cannot be employed in managing microbiome disruptions mediated by molecular signals from the enormous mass of microbial “dark matter,” given that an identified living bacterium is required as a target/host. 

Ultimately, in the context of the ecological definition of a microbiome, including not only bacterial taxa but also the products of the microbiota and the host environment, phage therapeutics can provide only partial solutions. 

Figure 5 shows a schematic summary of the pros and cons of bacteriophage therapy.

### 3.3. Taxon-Specific Metabolic Toxins

The conservation of molecular mechanisms in bacterial evolution provides an opening for another taxon-specific approach to targeted microbiome therapeutics, namely the administration of toxins that can “poison” specific bacterial taxa by exploiting common metabolic pathways not shared with other bacterial or human cells. 

For example, *Enterobacteriaceae*, frequent offenders in gut inflammatory processes, share molybdenum-cofactor-dependent respiratory pathways, which have been found to operate only during inflammatory episodes [61]. By using tungstate compounds to block these molybdoenzyme-dependent pathways, Winter, et al., have demonstrated selective “editing” of dysbiotic microbiota with concomitant reduction of intestinal inflammation and colonic tumors in mouse models of IBD and CRC [60,61,118,119,120].

Maini-Rekdal, et al. [121], used a similar approach in their work on bacterial modification of the Parkinson’s disease drug Levodopa (l-dopa). The metabolism of l-dopa in the gut reduces systemic drug availability and may contribute to untoward side effects of the drug in some patients, including potentially intolerable arrhythmias [121,122,123]. Findings that a first conversion of l-dopa to dopamine in the gut lumen by a resident *Enterococcus faecalis*-produced tyrosine decarboxylase enzyme is followed by further intralumenal modification of dopamine to a toxic metabolite, *m*-tyramine by a second organism, *Eggerthella lenta*, enabled discovery that (S)-afluoromethyltyrosine (AFMT), an L-tyrosine analogue, could selectively inhibit the formation of the toxic metabolite without damage to eukaryotic cells [121]. Here, the growth of neither bacterial species was affected by AFMT treatment, demonstrating that selective microbiome “editing” need not be directed at the modification of microbiome composition by taxa; instead, these studies amplify the notion that the molecular milieu itself can be favorably modified to alter phenotype.

Limitations of this approach include the requirement for new discovery work in identifying relevant bacterial pathways and identification of potential inhibitors in each new case, and the fact that such interventions can have no applicability in virally induced dysbiosis, where no metabolic activity by the pathobiont is involved. This approach shares the taxonomy-dependent limitations of phage therapeutics, having no capability to directly affect molecular mediators of dysbiosis on phenotype or to address the contributions of bacterial “dark matter” in dysbiosis.

Figure 6 shows a schematic summary of the pros and cons of taxon-specific metabolic toxin therapies.

## 4. Summary and Conclusions

In the several decades since culture-independent microbiology techniques have emerged, the science of “microbiomics” has grown explosively. Nonetheless, with only a few very recent exceptions, rigorously proven therapeutics for microbiome-associated diseases have emerged, and none are based on core fundamental principles of systems biology. With the exception of FMT, which is highly untargeted, other major attempts at selective modification of microbiomes in a generalizable way remain dependent on the taxonomic composition of dysbiotic microbiomes. Adherence to the taxonomy-driven view of dysbiosis, however, contributes to the “Anna Karenina Effect,” and arguably limits the full potential of therapies aimed at correcting dysbiosis and, ultimately, its protean manifestations across multiple disease processes.

A more successful approach seems likely to be one that is at least partially taxonomy-agnostic and fully systems-biology based, that is, one that recognizes the complexities—and promises—of recognizing the myriad molecular mediators that account for the phenotypic manifestations of dysbiosis. Such an approach will ultimately face fewer limitations in development of true microbiome drug discovery platforms and may open the possibilities of entire new classes and categories of therapeutics to fight today’s apparently insoluble chronic diseases.

## Figures and Tables

**Figure 1 pathogens-12-00238-f001:**
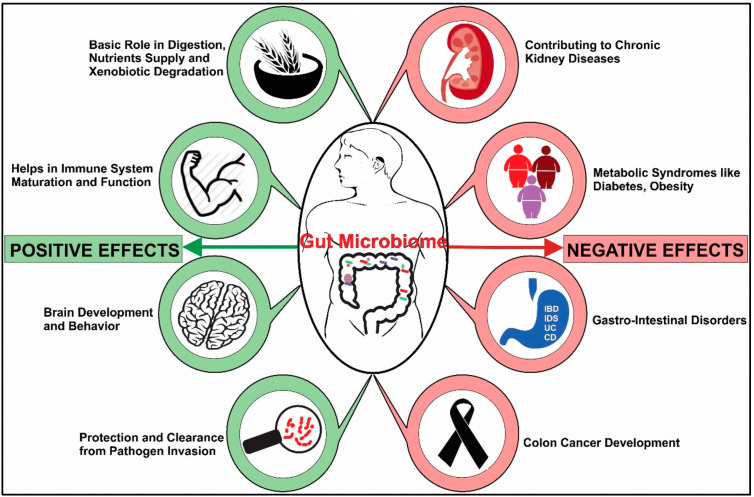
**Systems-wide Impacts of Dysbiosis.** General impact of microbiome status on function of the “host” or holobiont. Both positive and negative effects are mediated by bioactive molecules produced by bacteria, viruses, and fungi in the gut microbial ecosystem. Reproduced with permission from SNCSC from: Dekaboruah, E.; Suryavanshi, M.V.; Chettri, D.; Verma, A.K. Human microbiome: an academic update on human body site specific surveillance and its possible role. Archives of microbiology **2020**, 202, 2147–2167, doi:10.1007/s00203-020-01931-x [13].

**Figure 2 pathogens-12-00238-f002:**
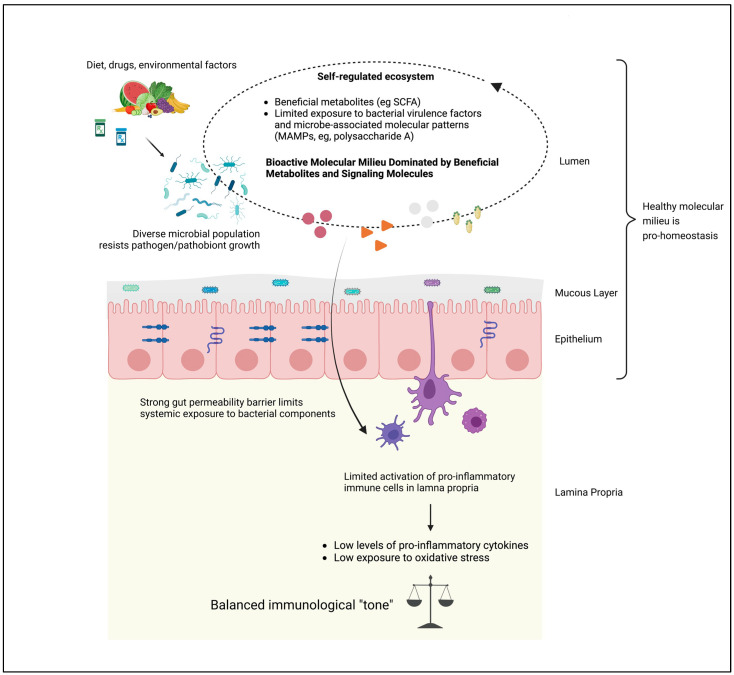
**Microbiome Ecosystem in Health.** In health, a self-regulated ecosystem forms, driven by diet and environmental factors that promote a diverse, pathogen-resistant microbiota, which in turn produces predominantly beneficial metabolites and limited exposure to deleterious microorganism-produced bioactive molecules. A normal mucous layer is maintained, and epithelial cell barrier function is sustained, limiting the amounts of lumenal materials translocated into the circulation and maintaining a low-inflammation, low oxidant-stress environment. Created with BioRender.com.

**Figure 3 pathogens-12-00238-f003:**
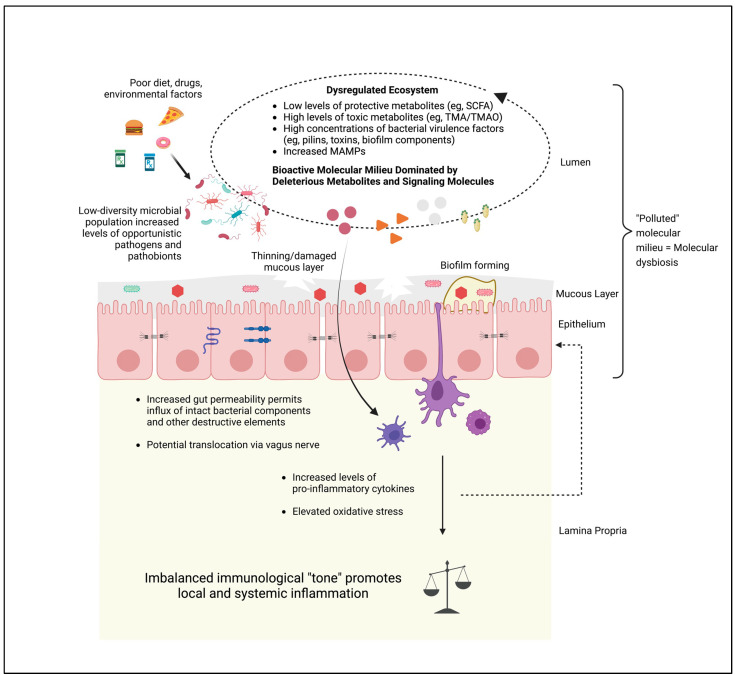
Microbiome Ecosystem in Dysbiosis. Dysbiosis is commonly triggered by dietary and other environmental factors, which differentially support opportunistic pathogens/pathobionts as microbial constituents. “Blooms” of pathobionts exclude beneficial bacteria from ecological niches, resulting in low-diversity populations dominated by bacteria that contribute to “molecular pollution” in the ecosystem. Such “pollution” consists of high levels of toxic or damaging bacterial metabolites, virulence factors, and microbe-associated molecular patterns (MAMPs). These disruptions contribute to a thinning mucous layer, support build-up of biofilms that can harbor and protect pathobionts, and damage epithelial junctional proteins resulting in influx of bioactive molecules into the epithelial layer and into the circulation, where they drive elevated levels of inflammation both within the gut and in extra-intestinal tissues. Created with BioRender.com.

**Figure 4 pathogens-12-00238-f004:**
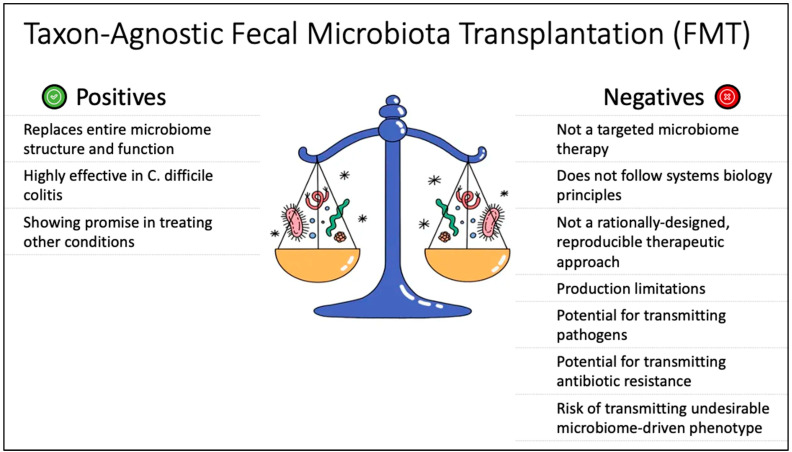
Schematic diagram of FMT—a taxon-agnostic, non-targeted approach to microbiome therapeutics, showing strengths and vulnerabilities compared with other modalities.

**Figure 5 pathogens-12-00238-f005:**
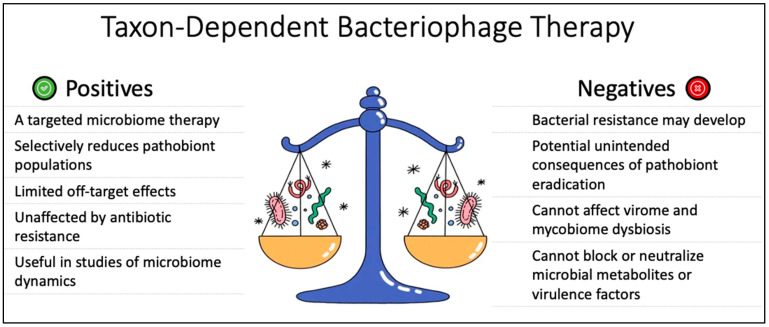
Schematic diagram of bacteriophage therapy, a taxon-dependent, targeted approach to microbiome therapeutics, showing strengths and vulnerabilities compared with other modalities.

**Figure 6 pathogens-12-00238-f006:**
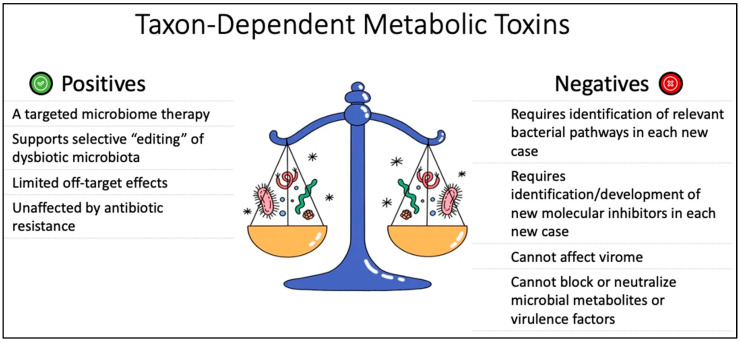
Shows a schematic summary of the pros and cons of taxon-dependent metabolic toxins, a taxon-dependent, targeted approach to microbiome therapeutics, showing strengths and vulnerabilities compared with other modalities.

## Data Availability

Data sharing not applicable. No new data were created or analyzed in this study. Data sharing is not applicable to this article.

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
