# Peer review of "A Taxonomy-Agnostic Approach to Targeted Microbiome Therapeutics—Leveraging Principles of Systems Biology"

_pathogens, 2023, doi:10.3390/pathogens12020238_

Round 1

Reviewer 1 Report

Authors Brumfield et al. have written a great review on the downsides of analyses that remain dependent on taxonomic classification. The manuscript is well written and well-conceived, and the authors present many compelling reasons to assume analytical approaches at the bioactive level. With a few minor revisions/additions, I recommend acceptance of this paper to Pathogens. Please see below:

1. Create an illustration for section 1 outlining what is discussed therein, namely the importance of the gut microbiome, and how interactions between the commensal biota and the host contribute to the health or dysbiosis of the holobiont. The authors seem to use BioRender; that would be a great platform to make this figure with. 

2. Change Figure 1A and 1B to Figure 1 and Figure 2, respectively. That will be easier to read. 

3. Create another figure summarizing the strengths and limitations of the therapeutic approaches discussed in Section 3 (i.e., FMT, phage therapy, and toxins). This would be good to visualize. 

4. Summary and conclusions should be section 4, not 3. Please correct. 

Upon the addition of the additional figures, and the other adjustments/corrections, I believe this paper will be of significant interest to the readers of Pathogens. 

Author Response

Thank you for this thoughtful and helpful review. We believe that we have addressed your concerns and incorporated your suggestions as follows:

Authors Brumfield et al. have written a great review on the downsides of analyses that remain dependent on taxonomic classification. The manuscript is well written and well-conceived, and the authors present many compelling reasons to assume analytical approaches at the bioactive level. With a few minor revisions/additions, I recommend acceptance of this paper to Pathogens. Please see below:

  1. Create an illustration for section 1 outlining what is discussed therein, namely the importance of the gut microbiome, and how interactions between the commensal biota and the host contribute to the health or dysbiosis of the holobiont. The authors seem to use BioRender; that would be a great platform to make this figure with. Thank you for this helpful suggestion. We have added a new Figure in Section 1. We believe this does add to the article’s clarity.
  2. Change Figure 1A and 1B to Figure 1 and Figure 2, respectively. That will be easier to read. Thank you – this is sensible, and we have separated the figures as Figures 2 and 3 of the revised manuscript.
  3. Create another figure summarizing the strengths and limitations of the therapeutic approaches discussed in Section 3 (i.e., FMT, phage therapy, and toxins). This would be good to visualize.  Thank you for this sensible suggestion. We have created three separate summary diagrams, placed at the end of each section (Figures 4, 5, and 6 of the revised text).
  4. Summary and conclusions should be section 4, not 3. Please correct. This has been done. Thank you!

Upon the addition of the additional figures, and the other adjustments/corrections, I believe this paper will be of significant interest to the readers of Pathogens. 

Reviewer 2 Report

This study approaches the rapid developments in the field of microbiome in the last 20 years from a different perspective. Very well written and structured. The Anna Karanina principle, which forms the basis of the study, will guide the understanding of the philosophy of the study. I have no additional comments to add except few minor points.

1- I understand that authors did not prefer to add "biotics" including probiotics and LTB in their context, it will be better to add some paragraphs about these interventions (to cover all). (No need to add this, just comment)

2- Lines 45-50: Authors describe basically the effects of microbiota. They could add some sentences about the gut microbiota axis.

3- Authors could also add some sentences about gut virome, gut mycobiome and their interactions with bacteriome.

4- Authors describe short and long term risk factors for microbiota. I think that they could add some phrases about the first 1000 days of life and microbiota composition (including pregnancy).

5- Line 60. I don' t prefer to use "flora" term in this very sophisticated article about microbiome and microbiome associated disoreders.

Author Response

Thank you for your very helpful review of our manuscript. We have incorporated most of your suggestions as follows:

This study approaches the rapid developments in the field of microbiome in the last 20 years from a different perspective. Very well written and structured. The Anna Karenina principle, which forms the basis of the study, will guide the understanding of the philosophy of the study. I have no additional comments to add except few minor points.

1- I understand that authors did not prefer to add "biotics" including probiotics and LTB in their context, it will be better to add some paragraphs about these interventions (to cover all). (No need to add this, just comment) Thank you for this comment – while we agree in principle, we believe that adding additional text would be unwieldy for this review. We appreciate that the Reviewer offers this as a comment and not a request.

2- Lines 45-50: Authors describe basically the effects of microbiota. They could add some sentences about the gut microbiota axis. Thank you for this suggestion. With the addition of the new Figure 1, we believe that we have addressed this concern – Figure 1 shows the various organ systems affected by the gut microbiome.

3- Authors could also add some sentences about gut virome, gut mycobiome and their interactions with bacteriome. Thank you for this observation. We have added text at lines 149-155, acknowledging these important interactions and pointing to future developments.

4- Authors describe short and long term risk factors for microbiota. I think that they could add some phrases about the first 1000 days of life and microbiota composition (including pregnancy).  Thank you for this comment. We have added text at Lines 97-104 to amplify these points in the context of the review.

5- Line 60. I don' t prefer to use "flora" term in this very sophisticated article about microbiome and microbiome associated disorders.  Thank you – we have changed this term to “microbial communities” now at line 60.